# Human Periapical Cyst-Derived Stem Cells Can Be A Smart “Lab-on-A-Cell” to Investigate Neurodegenerative Diseases and the Related Alteration of the Exosomes’ Content

**DOI:** 10.3390/brainsci9120358

**Published:** 2019-12-05

**Authors:** Marco Tatullo, Bruna Codispoti, Gianrico Spagnuolo, Barbara Zavan

**Affiliations:** 1Marelli Health, Tecnologica Research Institute, Stem Cell Unit, 88900 Crotone, Italy; bruna.codispoti@tecnologicasrl.com; 2Department of Therapeutic Dentistry, Sechenov University Russia, 19c1 Moscow, Russia; 3Department of Neurosciences, Reproductive and Odontostomatological Sciences, University of Naples, 80138 Napoli, Italy; 4Department of Medical Sciences, University of Ferrara, Via Fossato di Mortara 70, 44121 Ferrara, Italy; zvnbbr@unife.it

**Keywords:** Parkinson’s disease, human periapical cyst–mesenchymal stem cells, oral stem cells

## Abstract

Promising researches have demonstrated that the alteration of biological rhythms may be consistently linked to neurodegenerative pathologies. Parkinson’s disease (PD) has a multifactorial pathogenesis, involving both genetic and environmental and/or molecular co-factors. Generally, heterogeneous alterations in circadian rhythm (CR) are a typical finding in degenerative processes, such as cell aging and death. Although numerous genetic phenotypes have been discovered in the most common forms of PD, it seems that severe deficiencies in synaptic transmission and high vesicular recycling are frequently found in PD patients. Neuron-to-neuron interactions are often ensured by exosomes, a specific type of extracellular vesicle (EV). Neuron-derived exosomes may carry several active compounds, including miRNAs: Several studies have found that circulating miRNAs are closely associated with an atypical oscillation of circadian rhythm genes, and they are also involved in the regulation of clock genes, in animal models. In this context, a careful analysis of neural-differentiated Mesenchymal Stem Cells (MSCs) and the molecular and genetic characterization of their exosome content, both in healthy cells and in PD-induced cells, could be a strategic field of investigation for early diagnosis and better treatment of PD and similar neurodegenerative pathologies. A novel MSC population, called human periapical cyst–mesenchymal stem cells (hPCy–MSCs), has demonstrated that it naively expresswa the main neuronal markers, and may differentiate towards functional neurons. Therefore, hPCy–MSCs can be considered of particular interest for testing of in vitro strategies to treat neurological diseases. On the other hand, the limitations of using stem cells is an issue that leads researchers to perform experimental studies on the exosomes released by MCSs. Human periapical cyst-derived mesenkymal stem cells can be a smart “lab-on-a-cell” to investigate neurodegenerative diseases and the related exosomes’ content alteration.

## 1. Introduction

The human body works following specific rhythms, which constantly impact on several physiological activities and biological processes [1]. These rhythms can be slowly altered over time: This mechanism of resetting occurs in numerous living organisms, from bacteria to humans [2].

In the scientific literature, several biological rhythms have been described; among them, the most impacting and the most studied is undoubtedly the circadian clock, which depends on a great number of factors, such as the light/darkness rhythm, the environmental temperature, the oxidative stress, or the psychological condition [3]. 

Circadian genes actively regulate the circadian clock; such genes typically follow a sinusoidal regulation over the day and their expression can be fully understood after a daily round-the-clock analysis [4].

The synchronization of circadian rhythm is mediated by peripheral “secondary oscillators” working in several organs and following the primary information, transduced by cells in the central nervous system (CNS), via both direct and indirect pathways [5]. In more detail, direct synchronization is mediated by the humoral system, acting by the rhythmic releasing of hormones, such as cortisol; in this context, neural signals can directly impact on circadian synchronization, through the signals driven by the fibers of the autonomic nervous system [6]. On the other hand, the activity–rest cycles mediate the indirect entraining of peripheral oscillators; in fact, they are able to alter some important activities, like the feeding time and the secretion of the hormones [7].

Complex feedback loops allow clock genes to reciprocally regulate themselves. Clock mechanisms further influence the expression of circadian output genes, which strongly depend on circadian regulation [8]. Transcriptome profiling analyses in different tissues showed that up to 40% of genes are transcribed following a circadian regulation [9]. 

In the last years, clinical studies have consistently demonstrated that the alteration of circadian rhythms is prevalently related to pathologies linked to gastrointestinal dysmotility; such pathologies are also responsible for changes in gut microbiota composition [10]. Moreover, a new promising field of investigation tries to compare circadian rhythm alteration and neurodegenerative pathologies [11]. 

This article aims to describe the main cellular and molecular characteristics occurring in circadian rhythm alteration, taking into consideration the role of Mesenchymal Stem Cells (MSC)-derived exosomes in the pathogenesis of neurodegenerative diseases. Finally, the authors suggest how the exosomes derived from neural-differentiated human periapical cyst derived-mesenchymal stem cells (hPCy–MSCs) may represent a novel potential model for in vitro research on Parkinson’s disease (PD), to find new biomarkers, to test new drugs or, fatally, to find new pathways to be used in novel therapeutic approaches.

## 2. Factors Impacting PD Onset and Severity

People affected by PD may show a different phenotype, suggesting different genetic alterations among the causes of PD onset. Furthermore, specific genes involved in PD pathogenesis may indicate the type of development and the severity of the clinical symptomatology [12]. Genetic and environmental factors can modify the biology and the physiology of several biological processes; however, such risk factors seem to be differently able to influence PD severity [13]. 

In this regard, the link between PD and long-term exposure to outdoor air pollution was carefully investigated by the University of Utrecht, to assess whether the fine particles that pollute the air may have a role on PD severity and development. In this study, PD patients were recruited and compared to healthy patients: The results showed no significant difference in the risk of developing PD after long-term exposure to environmental fine dust [14]. Until a few years ago, the role of environmental factors in the development of PD was considered predominant; nowadays, new evidence suggests that the genetic component could play an important role. Over the past twenty years, link analysis and position cloning in large families with Mendelian forms of PD led to the identification of new genes involved the pathogenesis of the disease, while association studies have identified a growing number of loci and susceptibility genes responsible for sporadic forms of PD [15]. Numerous case-control association studies, both on candidate genes and the entire genome (Genome-Wide Association Study, GWAS), allowed the identification of genetic variants predisposing to PD in different populations. At the moment, four genes are known, clearly implicated in autosomal dominant Parkinson’s disease—the *SNCA*/*PARK1–PARK4* gene, the *LRRK2*/*PARK8* gene, the *VPS35*/*PARK17* gene, and the *EIF4G1*/*PARK18* gene; instead, *PARK-1* and *DJ-1* genes are involved in the autosomal recessive transmission disease. Currently it is not possible to establish the probability of inheriting this predisposition within a family. The rare exceptions are families in which the genetic predisposition is linked to a mutation of a single gene (monogenic disease) [16,17]. Oxidative stress is another interesting endogenous factor able to have a deep influence on PD onset [18]. In the last years, after the first studies on *PARKIN* gene alterations, an increasing number of genetic mutations have been tightly correlated to PD. Parkin was the first gene identified with Parkinson’s mutations. More than 100 mutations related to this pathology have been identified, point mutations, missense, truncation, deletion [19]. Parkin protein is codified by the *PARKIN* gene, and it is expressed in the brain, heart, and muscles; among all the PD-related proteins, Parkin is the most influenced by oxidative stress conditions. Aging may impact Parkin protein, as aging leads to cumulative oxidative stress that has been closely related to several cell damages, thus boosting the PD onset year-after-year. In this landscape, numerous studies have been carried out on aging and cell oxidation: Interestingly, concrete evidence has reported the presence of high levels of oxidative stress in the Substantia Nigra of PD patients. Dopaminergic neurons residing in the Substantia Nigra are mainly involved in PD pathogenesis, and they are also particularly vulnerable to oxidative stress; dopamine oxidation can generate toxic metabolites, such as dopamine–quinone, hydrogen–peroxide, and superoxide–radical, which are all converted into free radicals and reactive oxygen species (ROS) [20]. Generally, oxidative stress is an important cause of cellular damage, because of its effects on the peroxidation of membrane lipids: In PD neurons, oxidative stress causes an increased releasing of dopamine and the formation of super-aggregates of misfolded and toxic α-synuclein [21]. In vitro experiments have confirmed that in conditions of oxidative stress, the α-synuclein has a greater capacity to move from one cell to another, thus increasing the expression of the same protein. Using in vivo models, it has been shown that an over-expression of α-synuclein in cholinergic neurons of the dorsal motor nucleus of the vagus nerve (DMX) increases stress conditions with the accumulation of ROS, causing neurodegeneration [22]. 

Furthermore, three different missense mutations in α-synuclein have been identified as the cause of PD: p.A53T, p.A30P, and p.E46K [23,24]. In particular, subjects with this mutation have prominent dementia, observed about two years after the PD onset [24]. *LRRK2/PARK8* consists of 51 exons and codes for a protein of 2527 amino acids, called dardarin [25]. The *LRRK2* gene is frequently mutated in patients with PD. More in detail, the *G2019S* mutation is recognized as the most frequent genetic cause of Parkinson’s disease onset. Other mutations in the *LRRK2* gene are extremely rare. The clinical picture is essentially indistinguishable from idiopathic PD [26]. In a recent study, researchers have investigated the activity of *LRRK2*: It has been reported an activity six times higher in dopaminergic neurons of parkinsonian subjects, compared to control subjects. An impaired gene activity may interfere with the disposing of worn-out proteins, with the formation of α-synuclein protein accumulation [27]. *VPS35* and *EIF4G1* genes are still poorly investigated: The mutations in these genes may be a rare cause of PD, with a frequency of 0.1% and 0.02–0.2%, respectively. Differently from the dominant forms, such recessive transmissions are generally found in patients with a juvenile-onset of their pathology [28]. Finally, *DJ-1,* known as PARK7, is a pleiotropic protein belonging to the C56 peptidase family on chromosome-1 [29]. Mutations in the *DJ-1* gene (PARK7) are a rare cause of autosomal-recessive parkinsonism [30]. DJ-1 has multiple functions: It is prevalently involved in the regulation of antioxidant activities, working as a molecular chaperone [31]. The normal function of *DJ-1* and its role in dopaminergic cell degeneration is unknown, but there is evidence that links *DJ-1* to oxidative stress response and mitochondrial function. Canet-Aviles et al. have shown that, under oxidative stress conditions, the wildtype DJ-1 translocates to the outer mitochondrial membrane and is strongly related to better neuroprotection [32]. Thus, PD confirms its multifactorial pathogenesis, involving several genetic factors, but also requiring the contemporaneous presence of different environmental and/or cellular co-factors.

## 3. Circadian Rhythm Influences Biological Mechanisms Through Extracellular Vesicles Releasing

Exposure to sunlight can increase the release of the serotonin hormone. Serotonin plays a key role in regulating mood, sleep, and appetite: An alteration of the serotonin concentration consequently causes an alteration of the sleep–wake rhythm, which is a rhythm able to modify also the PD symptomatology [33].

A research team reported a study on patients carrying the Ala53Thr mutation of the α-synuclein gene; the enrolled patients showed a reduction in serotonergic transporters of 48%–57% in the nuclei of the brain base. According to this study, it is pretty clear that there is a correlation between the exposure to sunlight, the number of serotonergic transporters, the alteration of circadian rhythm, and PD symptomatology [34].

Circadian rhythm (CR) has been considered as a natural metronome, able to follow daytime, influencing several biological processes; in fact, the clock genes undergo circadian oscillation that reverberates in the main activities promoted by these genes. The molecular mechanism underlying the cyclic loop of clock gene expression in mammalian cells is still not fully understood; nevertheless, the transcriptional–translational feedback loops allow cells to start and stop a signaling process, then returning to baseline values [35].

The complex process regulated by CR is based on multiple regulatory feedback loops: The high complexity of this biological mechanism needs several levels of regulation, and the main checkpoints are located in transcriptional, post-transcriptional, and post-translational mechanisms. Interestingly, post-transcriptional regulation promoted by CR seems to modulate the production of rhythmic mRNA, as well as the expression of specific proteins [36].

The mammalian circadian rhythm is regulated by the continuous feedback from the transcriptional circuits. The *BMAL1*/*CLOCK* heterodimer guides the oscillating expression of such genes controlled by the promoters contained in the E-boxes [37].

In the last years, new information has been continuously shared on post-transcriptional regulation acting within the circadian system. Such regulation has been reported to impair the rhythm of both mRNA and protein expression [36]. The influence of transcription factors is known; on the contrary, the splicing factors that modulate this rhythm remain largely unknown [38].

The first study on post-transcriptional regulation of the circadian clock was based on the study of Lingulodinium [39]. Lingulodinium can show high bioluminescence, regulated by three components: The luciferin substrate, a luciferin-binding protein (LBP), and the luciferase enzyme (LCF) [40]. The clock controls the synthesis of these proteins, and both LBP and LCF are rhythmically expressed and reach their peak during the night. However, LPB (lipopolysaccharide binding protein) mRNA expression is not rhythmic, demonstrating that the expression of LBP and LCF is controlled by post-transcriptional pathways [41]. Further researches have shown that an RNA-binding protein, a clock-controlled translational regulator (CCTR), interacts with a repetition sequence UG in the 3′-UTR LBP and represses the LBP translation during the day [42]. McGlincy et al. have shown that the circadian clock regulates alternative splicing in the mouse. This experimental achievement reveals a new temporal dimension linked to the regulation of alternative splicing, and it also indicates that circadian biology is a complex system of factors depending on physiological pathways, alternative splicings, and other post-transcriptional regulatory mechanisms. These last discoveries have provided a platform for further studies on all those molecular mechanisms that regulate circadian exons and other post-transcriptional processes [43]. Current studies on mice, in fact, have shown that specific RNA-binding proteins involved in alternative splicings, such as *LARK*, *hnRNP* Q, *CIRP*, *4E*-Bp, are involved in the expression of core clock components by mediating the stabilization and mRNA translation mechanisms. However, the processes of co-transcription and post-transcription have been reported to be circadian-regulated in mammals, including AS, polyadenylation, mRNA stability, and their transport [44]. Other recent studies on Neurospora have confirmed the role of exosomes in CR regulation: The study has pointed out the strategic role of RRP44, a catalytic subunit of exosome and member of the RNase II family, in cell survival. The results demonstrated how RRP44 acts as a regulatory co-factor in the posttranscriptional negative feedback loop by regulating RNA degradation. Thus, similarly to what has been reported for Neurospora, the post-transcriptional negative feedback loops may be a mechanism working also in animal clocks [45]. Generally, heterogeneous alterations in CR are a typical finding in biological processes, such as cell aging and degeneration. Although numerous genetic phenotypes have been discovered in the most common forms of PD, it seems that mitochondrial dysfunction, altered lysosomal and proteasomal degradation, severe deficiencies in synaptic transmission, and high vesicular recycling seem to be found in several PD patients. Mitochondrial damages may increase the releasing of exosomes and the loss of Parkin in PD patients: They are the main molecular actors of the typical neuromuscular disorders affecting PD patients [46]. Exosomes are considered potential carriers of the toxic forms of α-synuclein, whose secretion could accelerate the PD progression [47,48]. It has been shown by Lonskaya et al. that in fibroblasts obtained from Parkin-deficient patients, there is an accelerated formation of ILV (intraluminal vesicles), which will become exosomes, and an increase in the number of intraluminal membranes, thus causing an increase in the releasing of exosomes. This mechanism is triggered by the fact that Parkin normally regulates the organization of the tubular regions that influence exosomal secretion [49].

Moreover, the increased release of exosomes in Parkin-deficient cells could be related to the secretion of α-synuclein [46]. *PARKIN* and *PINK1* have been found in many intracellular organelles of PD patients, and they can impact several mitochondrial activities. A recent study carried on a Drosophila Melanogaster model has demonstrated that mutant forms of such proteins can be associated with the absence of circadian locomotor anticipatory activity in the morning, and with altered sleep, compared to controls [50,51].

The exosomes secreted from neurons can carry and release several molecules; among these molecules, miRNAs have a specific role in the regulation of circadian rhythm. 

Some circulating miRNAs, such as miRNA-152, miRNA-494, and miRNA-142-3p, are linked to the regulation of the circadian rhythm; in particular, such miRNAs are linked to the diurnal oscillation and to the setting of clock genes, as demonstrated in murine models. In more detail, miRNA-494 and miRNA-142-3p can regulate the circadian rhythm through post-transcriptional regulation of the clock gene *BMAL1* [52].

Due to the central role of exosomes in both healthy and pathological brain, it is reasonable that exosomes may play an important role in the pathogenesis of mental disorders, such as Parkinson’s and Alzheimer’s diseases. 

It has been demonstrated that exosomes play a significant role in the mechanisms involved in the pathogenesis of several mental disorders, such as neuroinflammation and epigenetic regulation [53].

MiRNAs have been linked to some mental disorders; thus, it is reasonable that the miRNAs contained in exosomes could actively contribute to the progression of pathological forms, as well as to the onset of individual symptoms in mental disorders.

There is mounting evidence about the role of exosomes in neuroinflammation; furthermore, the signaling between neurons and glial cells is also mediated via exosomes. This cell-to-cell communication represents a mechanism able to promote synaptic plasticity, thanks to active synapses that stimulate the activation of such inactive synapses.

Exosomes are able to cross the blood–brain barrier; this important ability makes them good biomarkers for neural diseases. Particularly interesting is the possibility to characterize exosomes based on their cellular origin; this fact potentially allows the knowledge of important information about the disease to be investigated [53]. 

## 4. The Human Periapical Cyst–Mesenchymal Stem Cells (hPCy–MSCs) in Regenerative Medicine

Mesenchymal stem cells (MSCs) are massively located in the bone marrow, but they are also somewhat present in each tissue of the human organism [54]. MSCs hold remarkable proliferative and differentiating abilities, as well as great plasticity: They can typically differentiate in the three classic stromal sheets, such as fat, bone, and cartilage [55].

Periapical cysts develop due to the chronicity of inflammatory conditions involving the apical area of the tooth, following the onset of endodontic infections. Maeda and colleagues in 2004 noticed the onset of new bone formation in the periosteum surrounding the area of excision of an inflammatory cyst; this observation put in place the hypothesis of the presence of osteogenic progenitors in the removed granulation tissue [56].

In 2013, a novel MSC population was isolated from the inner wall of the dental periapical inflammatory cyst that typically develops as a result of necrosis of the dental pulp: The human periapical cyst–mesenchymal stem cells (hPCy–MSCs) [57]. hPCy–MSCs were widely investigated, and they show no hematopoietic marker, such as CD45, CD34, or HLA-DR [16,17]. Immunofluorescence and cytofluorimetry analyses on hPCy–MSCs have demonstrated that they naively express the main neuronal markers, such as β-III tubulin, and the main astrocytes markers, such as glial fibrillary acidic protein (GFAP). Using q-PCR studies, the presence of transcripts was identified for genes associated with the neuronal phenotype, such as β-III tubulin, and for genes associated with the dopaminergic phenotype, such as tyrosine hydroxylase [58]. Gene expression and levels of protein production in hPCy–MSCs exposed to neurogenic culturing conditions showed a notable increase in neural markers [59]. 

Recently, several researches have been performed, taking into consideration the reusing of biological wastes [60,61]. The scientific community has consistently demonstrated the safety of biological wastes, if they are properly treated and manipulated, in clinical applications and cell-banking [62].

Bone marrow and peripheral blood typically represent the most recognized sources of stem cells for tissue repairing; however, MSC isolation from such sources is quite invasive, and the amount of stem cells harvested is often poor [63]. Therefore, hPCy–MSCs can be considered as a promising source of immature cells, useful for various clinical applications, with particular regard to bone regeneration and the treatment of neurological diseases [64].

The possibility of obtaining mesenchymal stem cells from a “biological waste” creates the conditions to exploit an “alternative” source of MSCs, at zero biological cost. hPCy–MSCs are obtained from an alternative source that completely encompasses the concept of modern waste medicine: The regeneration of any tissue can be achieved by differentiating these MSCs. Particularly, the osteogenic and neurogenic differentiations have been obtained with hPCy–MSCs, showing a marked commitment and creating interesting perspectives for their use in neurodegenerative diseases [55]. hPCy–MSCs are easily obtained from oral tissues: A pretty accessible source of stem cells is the body fat. Mesenchymal stem cells derived from adipose tissue (ADSCs) are adult stem cells extracted directly from body fat. ADSCs have been investigated and clinically used: ADSCs are able to stimulate the production of bone and cartilage. Furthermore, ADSCs have a remarkable proliferative capacity, both in vivo and in vitro, preserving a multilineage potential. ADSCs are able to show a specific ability to secrete growth factors, inducing angiogenesis, such as vascular endothelial growth factor (VEGF), platelet-derived growth factor (PDGF), and hepatocyte growth factor (HGF). Moreover, ADSCs have been reported to have a high expression of insulin-like growth factor-1 (IGF-1), vascular endothelial growth factor-D (VEGF-D), and interleukin-8 (IL-8), improving the proangiogenic commitments of ADSCs [65]. The use of stem cells from adipose tissue is very promising in clinical practice; however, the use of hPCy–MSCs is based on MSCs from wasted tissues, thus reducing the biological impact. The real challenge is to translate the in vitro results into an in vivo context, and therefore, future investigations should be conducted to confirm the regenerative potential of these MSCs. On the other hand, the limitations in using stem cells for tissue repairing have pushed researchers to find alternative solutions to achieve good results with low ethical and biological issues. The solution seems to come from the use of exosomes; in fact, exosomes are carriers of important molecules that often act as messengers from cells to other cells, thus creating continuous crosstalks able to influence the activity of the local environment [66]. 

The role of exosomes is important from several different points of view; in fact, exosomes may be important study models to investigate the correlation among their content and the presence/evolution of a specific pathology. On the other hands, exosomes’ content may be an important target for future drug therapies to develop [67].

In this landscape, the levels of α-synuclein can be considered highly pathognomonic in the pathogenesis of PD. Interesting studies were aimed to quantify α-synuclein at the plasma level and in the neural-derived exosomes, in both healthy and PD patients [68]. The results of such studies highlight how the plasma levels of α-synuclein were substantially unchanged between the two groups, while the protein levels in the neural-derived exosomes were significantly higher in subjects with PD. A similar role has also been observed by the protein Deglycase DJ-1, also known as Parkinson disease protein 7: High DJ-1 and α-synuclein levels were observed in neural-derived exosomes obtained by patients with PD. The interaction between such two proteins, and their increased presence in the neural-derived exosomes of PD patients, has not been completely clarified, but it cannot be ignored in a future approach on early diagnosis and therapy of PD [69]. Abnormal α-synuclein aggregation can be thus considered as the hallmark of PD [70].

Besides, there is growing literature confirming that α-synuclein could also be a co-factor in the development of several psychiatric disorders. This link among neurodegenerative and psychiatric diseases may recognize some common factors, such as the dysregulation of the cytokine system, and the alteration of circadian rhythm [71]. hPCy–MSC-derived exosomes can be considered as a “concept lab-on-cell”: in this new lab’s concept, we can investigate novel therapies based on extracellular vesicles and their content [23]. To achieve reliable results, a major issue is represented by the proper isolation and characterization of exosomes [72]. Indeed, exosomes are represented by several types of vesicles, classified by their size and their molecular pattern that make them able to overcome pored filters and to be selected by means of high-speed centrifugation [73]. Current procedures for exosome isolation allow us to obtain a heterogeneous population, which does not represent the proper research medium, as the different types of vesicles can certainly alter the final result of any investigation made on their content. In this landscape, several researchers have recently tried to carry out a protocol able to isolate different populations of exosomes, by means of high-speed ultracentrifugation and filtration [74]. The extracellular vesicles (EVs), properly isolated, will be used for studies on mitochondrial components and molecular mediators showing a role in the functional decay of mitochondria and cells of PD patients [75].

## 5. Conclusions and Future Insights

Circadian rhythm is closely related to several cognitive impairments, suggesting its possible use as a biomarker for such circadian alterations highlighted in PD patients [76]. The investigation of different non-coding RNAs detected in neurons of PD patients may lead to promising strategies to modulate symptoms and conditions associated with neurodegenerative disorders. Several neurodegenerative pathways are closely related to autophagy and apoptosis. Circadian clock genes are necessary to modulate autophagy, limit cognitive loss, and prevent neuronal injury. On the other hand, non-coding RNAs seem to control neuronal stem cell development and neuronal differentiation [77]. Neuron-derived exosomes contain several active compounds, including miRNAs [78]: Several studies have found that circulating miRNAs are closely associated with circadian rhythm gene oscillation, and they participate in the regulation of clock genes in animal models [52].

The searching for PD-related early markers in cell culture media is obtained by comparing these novel markers with the ones typically present in biological fluids [53]. This research could be carried out by analyzing different pools of proteins, or some specific RNA, present in the exosomes released in the culture medium of cells obtained from patients at the onset of the PD [79].

hPCy–MSCs have demonstrated that they can differentiate into functional neurons [80]: A selective death of dopaminergic neurons characterizes the long pre-symptomatic phase of PD, thus, a careful analysis of neural-differentiated hPCy–MSCs and the molecular and genetic characterization of their exosome content, both in healthy cells and in PD-induced cells, will be a strategic field of investigation for early diagnosis and better treatment of PD and similar neurodegenerative pathologies. It is important to characterize the biochemical and molecular targets that may anticipate or worsen the clinical onset of PD: A proper understanding of preliminary signals can characterize specific therapies, independently from the genetic or environmental pathogenesis.

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
