# Peer review of "Human Periapical Cyst-Derived Stem Cells Can Be A Smart “Lab-on-A-Cell” to Investigate Neurodegenerative Diseases and the Related Alteration of the Exosomes’ Content"

_brainsci, 2019, doi:10.3390/brainsci9120358_

Round 1

Reviewer 1 Report

Tatullo et al. described the main cellular and molecular characteristics occurring in circadian rhythm (CR) alteration. Generally, CR abnormalities are a typical finding in neurodegenerative disorders such as Parkinson’s disease (PD). Authors taking into consideration  the possibility to study the role of mesenchymal stem cells (MSCs)-derived exosomes in the pathogenesis of PD. In particular, they focalized on the MSCs isolated from human periapical cyst (hPCy-MSCs) given that they naively express the main neuronal and astrocyte markers. Finally, authors suggested how the exosomes derived from neural-differentiated hPCy-MSCs may represent a novel potential model for the in vitro research on PD.

Although authors highlight the proposed issues in a comprehensive way a minor issue have to be addressed before considering it suitable for publication.

Most references do not match the text. Authors have to correct the following mistakes:

Reference 15 do not match the study of Maeda and colleagues (2004). References from 18 to 22 do not match the text. References 24-25 do not match the text (protocol able to isolate different population of exosomes). Reference 26-31 and 32-38 do not match the text. Moreover, some of these references do not match topics covered in the manuscript. Authors have to report at least a reference for the sentence on page 3 lines 141-144 (Mesenchymal stem cells…)

Author Response

Dear Reviewer n.1, thank you so much for your positive comments.

Please, see the enclosed letter for further details on our revised manuscript.

Best wishes,

Marco Tatullo, on behalf of co-Authors

Reviewer 2 Report

In the paper named” Human Periapical Cyst-derived stem cells Can Be A  Smart “Lab-on-A-Cell” to Investigate on Neurodegenerative Diseases and the Related Alteration of the Exosomes’ content” the authors emphasize on the analysis of exosomes’ contents to diagnose the early stage of neurodegenerative diseases. As a strategy towards it, they discussed the human Periapical-Cyst-Mesenchymal Stem Cells (hPCy-MSCs) as they express the main neuronal markers and it may help to differentiate functional neurons. In this perspective, they initially mentioned about the effect of environmental and genetic effect, Circadian rhythm influences biological mechanisms, and then the application of human Periapical-Cyst-Mesenchymal Stem Cells (hPCy-MSCs) in regenerative medicine. Although this perspective is focusing on the most interesting trend in regenerative medicine for the neurodegeneration disease, the information in this manuscript is not adequate and requires some improvements.

In the environmental and genetic factors, authors mentioned about air pollution and oxidative stress but information regarding genetic factor require more attention with more discussion and their consequence.

In section 3 regarding circadian circle, probably discussion of alternative slicing and their consequence on the RNA content as well as RNA degradation during cell development would make it more interesting to the audience.

In section 4, I am wondering if authors could introduce a pictorial presentation presenting the ongoing model or hypothesis around the human Periapical-Cyst-Mesenchymal Stem Cells (hPCy-MSCs) and neurodegeneration disease.

Additionally, authors can incorporate more references in their manuscript in the different sections in this manuscript.  

Author Response

Dear Reviewer n.2, thank you so much for your positive comments.

Please, see the enclosed letter for further details on our revised manuscript.

Best wishes,

Marco Tatullo, on behalf of co-Authors

Reviewer 3 Report

The authors present a Perspective on the usage of human periapical cyst-derived mesenchimal stem cells (hPCy-MSCs) as a new source of immature neural cells that could be potentially used as a cellular model in which to investigate the molecular mechanisms underlying the pathogenesis of PD and to find new biomarkers, for example RNAs and proteins in the exosomes released by these cells, of early PD. 

To focus in these cells is certainly innovative and original, and it might be of interest to the readers, however, the ideas are not well presented along the text and the message is not delivered in a clear and comprehensive way. The "conclusions and future insights" section is probably the best part, where a series of ideas are presented in a rational order and where the flow or the arguments can be followed in a more or less clear way. The rest of the text should be carefully revised and re-written.

The section about factors impacting PD onset and severity is not comprehensive and does not summarize the key findings on this topic. In addition, references to the original articles for most statements are missing.

The section about circadian rhythms is also confusing and the different arguments for linking circadian alterations with PD, and with the detection of these alterations in exosomes, are not clearly exposed. In addition, the articles referred to in lines 132-134 do not support the "increased releasing exosomes" statement. 

Finally, the section about hPCy-MSCs provides more information but still the key points of the advantages of using these cells compared with other sources of stem cells is not written in a clear and concise way. In addition, key references are missing to support some of the statements (examples: line 142, line 180). Finally, references 24 and 25 in line 203 do not refer to the aspect described in the corresponding sentence.

Author Response

Dear Reviewer n.3, thank you so much for your positive comments.

Please, see the enclosed letter for further details on our revised manuscript.

Best wishes,

Marco Tatullo, on behalf of co-Authors

Round 2

Reviewer 2 Report

No comment.

Author Response

We deeply thank the reviewer n.2 for this final acceptance.

Reviewer 3 Report

The authors have worked hard on the revised version of the manuscript and the article has definitely improved. The idees and more clearly presented, better organised and the flow of the text has significantly improved.

I still have some concerns about the way authors touch upon the role of miRNAs. The importance of those is highlighted in the abstract and in the conclusions but it is not even mentioned in the main body of the perspective. The authors should introduce the importance of miRNAs in the main text.

Author Response

Dear Reviewer n.3,

We really thank you for your valuable suggestion, properly addressed to further improve the quality of our manuscript.

In the section 3, a better explanation of the role of miRNA, in the pathways related to the circadian rhythm and in the main neural diseases, has been added.

We feel that article has been definitely improved.

We hope the article will be accepted.